# Population Dynamics of *Drosophila suzukii* in Coastal and Mainland Sweet Cherry Orchards of Greece

**DOI:** 10.3390/insects11090621

**Published:** 2020-09-10

**Authors:** Stella A. Papanastasiou, Vasilis G. Rodovitis, Evmorfia P. Bataka, Eleni Verykouki, Nikos T. Papadopoulos

**Affiliations:** Department of Agriculture, Crop Production and Rural Environment, University of Thessaly, Fytokou St., 384 46 Volos, Greece; spapanast@uth.gr (S.A.P.); rodoviti@uth.gr (V.G.R.); bataka@uth.gr (E.P.B.); e.verykouki@gmail.com (E.V.)

**Keywords:** spotted wing drosophila, invasive pest, adult phenology, flight activity, sweet-cherry susceptibility, infestation levels

## Abstract

**Simple Summary:**

*Drosophila suzukii* or spotted-wing drosophila is a fruit fly of the same family as the vinegar fly. The majority of Drosophila fruit flies are not considered agricultural pests. However, *D. suzukii* is a pest of several high-value soft and thin skin fruits such as strawberries and sweet cherries. *Drosophila suzukii* is also considered worldwide as one of the most important invasive insect pests. We monitored the insect’s flight and we assessed the infestation levels of different sweet-cherry cultivars, in coastal and mainland cherry orchards of Greece, for two consecutive years (2018–2020). Adults were captured throughout the year in the coastal area with two peaks in spring and late-autumn. Captures were nearly zero during the hot summer months. Trap captures exhibited only one peak in autumn at the mainland area, and ceased during winter and spring. Higher sweet-cherry infestation levels were recorded in the coastal than in the mainland area and in unmanaged than in commercial orchards. Both early and late-ripening cultivars were highly susceptible to *D. suzukii* infestation in the coastal area. Infestation rates were higher in late-ripening cultivars than in early-ripening ones in the mainland area. We conclude that *D. suzukii* has adapted well to the Mediterranean climate of Greece, and is able to progressively exploit several crops and wild hosts of mainland and coastal areas.

**Abstract:**

Despite the recent invasion and wide spread of *Drosophila suzukii* Matsumura (Diptera: Drosophilidae) in Europe, little is known regarding its population trends in coastal areas of the southern Mediterranean countries. Using adult trapping and fruit sampling, we studied the population dynamics of *D. suzukii* in coastal and mainland (semi-highland) cherry orchards of Greece, from 2018 to 2020. Adults were captured in traps baited with apple cider vinegar, placed in conventional and unmanaged sweet-cherry orchards, and in neighbouring wild growing hosts. Sampling of sweet-cherry fruit to assess infestation levels was conducted from early and late-ripening cherry cultivars in both areas. Adults were captured throughout the year in the coastal area with two peaks registered in spring and late-autumn. Captures were nearly zero during the hot summer months. Flight activity exhibited only one peak in autumn at the mainland area, and ceased during winter and spring. Captures in wild hosts were lower during the sweet-cherry ripening period than later in the season. Higher sweet-cherry infestation levels were recorded in the coastal than in the mainland area and in unmanaged than in commercial orchards. Both early and late-ripening cultivars were highly susceptible to *D. suzukii* infestation in the coastal area. Infestation rates were higher in late-ripening cultivars than in early-ripening ones in the mainland area. We conclude that *D. suzukii* has well adapted to the Mediterranean climate of Greece, and is able to progressively exploit several crops and wild hosts of mainland and coastal areas.

## 1. Introduction

The Spotted Wing Drosophila (SWD), *Drosophila suzukii* Matsumura (Diptera: Drosophilidae) is a pest of several high-value soft fruit and fruit with thin epicarp, such as blackberries and raspberries—*Rubus* sp., blueberries—*Vaccinium* sp., strawberries—*Fragaria* × *ananassa* and sweet cherries—*Prunus avium* [1,2,3]. It also infests a wide range of wild and ornamental plants which support high fly densities alongside cultivated areas [4,5,6]. Unlike sibling drosophilid species, which cause secondary infestations in overripe, wounded or semi-rotten fruit, the *D. suzukii* females possess a serrated ovipositor which enables them to penetrate and oviposit intact ripening or ripe fruit [7]. Larval feeding within the mesocarp leads to fruit deterioration, promotes secondary bacterial and/or fungal infestations and increases the fruit susceptibility to damage caused by other Drosophilidae [8,9]. Polyphagy, multivoltinism and high reproductive potential [10] of this insect species lead to rapid population growth, fruit damage and great economic losses in commercial small fruit orchards [11].

*Drosophila suzukii* is currently considered worldwide as one of the most important invasive insect pests of soft fruits [12]. Native to southeastern Asia, it has invaded and established, in the last decade, other Asian regions, as well as Europe and the Americas [13]. Environmentally adequate areas with potential for *D. suzukii* occurrence exist in Oceania and Africa, although there has been no record of its presence on these continents [14]. Although the first record of *D. suzukii* as a damage-causing fruit-pest was in 1939 in Japan [15], reports of severe losses of soft fruits were reported much later in Europe (2009), north America (2008) and south America (2013) [13,15,16,17]. In Europe, the fly was detected simultaneously in several Mediterranean countries (Spain, France and Italy) during 2008–2009 [18]. Subsequently, from 2010 until 2016 the presence of *D. suzukii* in almost all European countries was confirmed (reviewed in [17,19]). In Greece, *D. suzukii* was first recorded in August 2013, in blackberries and raspberries at the prefecture of Epirus (northwestern Greece—Ioannina) [20]. The following spring the fly was also trapped at the southeastern shore of the Crete island (southern part of the country) [21,22]. Although *D. suzukii* appears to be currently present in almost all parts of the country, official data regarding distribution and establishment in Greece are rather scarce [23].

Sweet-cherry cultivation is expanding in Greece and is currently one of the main domestic fruit crops. The national sweet-cherry production ranks 12th globally, and 6th among other European countries. The total cultivated and harvested area almost doubled from 1990 (8500 ha) to 2018 (16,210 ha) [24]. Likewise, sweet-cherry production indices followed an increasing trend reaching 70,000 tonnes in 2014, 83,000 tonnes in 2016 and 90,000 tonnes in 2018 [24]. Although sweet-cherry orchards can be found throughout most of the agricultural areas of the country, 90–95% of the cherry producing areas are located in highland or semi-highland regions of northern (Macedonia), central (Thessaly) and southern (Peloponnese) Greece. Given that sweet-cherry production is at the moment among the most proliferous domestic crops, yielding USD 30–50 M annually from exports, more farmers are willing to invest in sweet-cherry cultivation. *Drosophila suzukii* has relatively recently invaded and established in Greece, resulting in the collection of scarce information regarding the insect’s range expansion in northern and central fruit producing areas, such as Pella and Magnisia. Moreover, no information exists concerning the infestation risk and the susceptibility levels of several sweet-cherry cultivars, in relation to the different areas of cherry cultivation.

Spotted Wing Drosophila adults are highly active, migrating from cultivated crops to unmanaged and wild host plants during the fruiting season and from low to high altitudes during seasonal climate alterations in temperate regions [25,26]. Seasonal host availability and suitable climatic conditions in a given area may largely support high population densities of this pest that are difficult to control. Chemical applications target reproductively mature females before ovipositing in susceptible fruit. Overlapping generations (multivoltinism), continuous availability of hosts and migration of *D. suzukii* adults lead to dramatic population increases and require repeated insecticide applications to manage them. For example, cherry growers in North America follow a calendar-based, weekly spraying of synthetic insecticides starting on the onset of cherry-ripening and ending before harvest [27]. Successive insecticide applications are uneconomical, can induce resistance to several compounds and threaten non-target organisms and human health. Data on population dynamics in a given area are essential in order to design an adequate pest management plan that may include fruit sanitation practices and a strategic application of chemical control.

Patterns of the phenology of adult *D. suzukii* in different cherry producing areas may vary tremendously depending on the geographic location, and several other factors such climatic particularities, cherry tree phenology, availability of alternative hosts, etc. For example, adult captures of *D. suzukii* in cherry producing areas of central California show two peaks, one in spring and another in fall, while high migration rates among available hosts (wild, cultivated) following cherry harvest is recorded [26]. On the other hand, trap captures in cherry producing areas of Oregon peak from August to November, well after cherry harvesting [27]. In addition, the levels of cherry infestation and peak trap captures do not coincide in areas of California with mild winter conditions. This indicates that trapping data should not be uniquely considered when designing a pest management program. It seems that population dynamics are strongly related to the climate and cherry tree phenology at a local scale. The sweet-cherry industry in Greece is currently not fully aware of the severity of the *D. suzukii* problem. The majority of cherry farmers fails to identify infestations by *D. suzukii* and usually attribute them to *Rhagoletis cerasi* (L.) (Diptera: Tephritidae), which traditionally infests sweet-cherries throughout Greece. Apart from personal communications with cherry producers and scarce reporting of unexpected sweet-cherry fruit deterioration close to harvest by unidentified pests, nothing is known regarding the insect’s population dynamics in cherry producing areas of Greece.

The present study investigates for the first time the population dynamics of *D. suzukii* adults in two geographically distant cherry producing areas of the Mediterranean region, with different climates and landscapes. We also report infestation levels of different sweet-cherry cultivars. The specific aims were to determine (a) the seasonal patterns of *D. suzukii* in a coastal and a mainland (semi-highland) cherry producing area of Greece, (b) the population dynamics in conventional and unmanaged cherry orchards, and in neighboring wild hosts, (c) the susceptibility of early and late-ripening sweet-cherry varieties in the two areas.

## 2. Materials and Methods

### 2.1. Field Sites and Trap Allocation

Our study was conducted in two cherry producing areas of Greece: Agia Fotini (Pella County, semi-highland area, northern Greece) and Lehonia (Magnisia County, coastal area, central Greece). Pella County is among the top cherry-producing areas of Greece. Although a large part of this region consists of mountains (45%), two main valleys (Aridea and Giannitsa) and a wealth in rivers and ground-water reserves contribute to the area’s agricultural production. More than 31,000 ha are fruit tree orchards with most of them dedicated to stone fruit crops (e.g., peaches, nectarines, plums and cherries). Sweet-cherry orchards are located at the semi-highland areas of the County, often neighboring with forest and wild bush vegetation, including a recent expansion of sweet-cherry cultivation in the valleys (10,500 ha). The rest of the orchards (peaches, nectarines) are located in the two main valleys. Magnisia County has a rather diverse geography with Pelion Mountain meeting the Aegean Sea to the East and the coastal fertile plain of Lehonia to the West. A total of 17,000 ha in Pelion and the neighboring plain are olive groves and 320 ha are cultivated with sweet-cherries, plums and citrus.

In order to monitor the flight activity of *D. suzukii* adults, we placed traps in two cherry orchards of Agia Fotini during 2018, as well as on unmanaged cherry trees and on wild vegetation alongside the two orchards, mainly consisting of wild *Rubus* sp. In 2019, traps were placed in four more cherry orchards and on wild vegetation of the same area. In Lehonia, traps were placed during spring of 2018 in four different orchards (1: abandoned/unmanaged mixed orchard, 2: conventional cherry orchard, 3: conventional mixed orchard and 4: organic mixed farm with fruit trees and vegetables). Both organic and conventional mixed orchards included apple, pear, fig, peach, apricot and quince trees (Table 1).

The traps used were made with transparent plastic bottles (6 cm base diameter, 3 cm screwing cap diameter, 20 cm height, 0.5 Lt volume) where six lateral holes (0.4 cm diameter) were drilled at the upper part (5 cm from the bottle top). Pipette tips (200 μLt volume) were cut at 0.5 cm of the edge and adjusted inside the bottle holes in order to prevent the exit of *D. suzukii* adults. Traps were lured with 150 mL of apple cider vinegar, apple juice and sugar (3:1:0.1) [4,28], and were placed at a height of approximately 1.5 m on cherry trees and on nearby wild hosts. The content of the traps was collected on a weekly basis and the attractant was refreshed every other week. Adult *D. suzukii* were separated and counted by sex under a stereomicroscope.

### 2.2. Climatic Data

Climatic data (mean daily temperature and precipitation) for the two experimental areas were obtained from a national database (http://meteosearch.meteo.gr/). Two weather stations that were more proximal to the experimental areas of our study were located in Volos (6.5 km north-west of Lehonia) and in Agios Pavlos (3.5 km south-west of Agia Fotini). Mean daily temperatures and precipitation for the two areas from March 2018 to March 2020 are given in Figure 1 and Figure 2, respectively.

### 2.3. Sweet-Cherry Fruit Sampling

During the harvest periods of 2018 and 2019, in Lehonia and Agia Fotini, 100–300 sweet cherries were sampled from early- and late-maturing cherry cultivars to assess the infestation levels from *D. suzukii*. Moreover, 100 sweet-cherry fruits were sampled from untreated, early- and late-maturing cherry cultivars of both locations (Table 2). Fruits were transferred to the Laboratory of Entomology and Agricultural Zoology, University of Thessaly, were placed individually in small plastic containers (5 cm base diameter, 5 cm height) and kept at constant conditions (25 °C ± 2 °C, 65 ± 5% R.H., L14:D10 photoperiod). Each container had an opening, on the upper part, covered with fine-mesh organdie to provide proper ventilation and prevent escape of *D. suzukii* adults. Each fruit was placed on a cotton disc to absorb exuding liquids of mesocarp deterioration due to larval feeding. Cherries were inspected twice a week for two consecutive weeks, and once all adults emerged, they were counted under a stereoscope. Each cherry was also inspected for any additional pupae that did not yield adults. The number of infested cherries and that of *D. suzukii* adults was recorded for all treatments.

### 2.4. Statistical Analysis

The mean number of flies captured in each trap for every observation date (Appendix A), as well as the sum of adults captured per trap per day (Flies/Trap/Day: FTD), were calculated for the whole period of flight activity and were used at the graphic depiction of population dynamics in the two geographic locations and among different treatments (Appendix A). FTDs were calculated overall for the two geographic areas (Lehonia and Agia Fotini), as well as for the different treatments within each geographic area for the two consecutive years (20 March 2018 to 22 March 2020) of flight monitoring. To better depict the insect’s annual population cycle [29], trap capture patterns were also presented from March 20 to March 19 of the following year for the two consecutive years (see Appendix A).

Flight monitoring data were analyzed with R v4.0.0 (R Core Team 2013,R Foundation of Statistical Computing, Vienna, Austria) in RStudio v1.1.453 (RStudio 2012, R Foundation of Statistical Computing, Vienna, Austria) by using the “geepack” package [30,31,32] which estimates the parameters of a generalized linear model using generalized estimating equations (GEEs), in order to account for the within-subject dependency. Moreover, SPSS 26.0 (SPSS Inc., Chicago, IL, USA) was used for the analysis of fruit infestation data.

We used GEEs to compare the numbers of adult *D. suzukii* captured between the two geographic locations (Lehonia, Agia Fotini), across different seasons (spring, summer, autumn, winter) and between the two years of flight monitoring (20 March 2018 until 19 March 2019 and 20 March 2019 until 22 March 2020), including the mean daily temperature per location as a covariate. Using GEEs, we also evaluated the trap captures among different treatments (conventional cherry orchards, unmanaged cherry trees, mixed organic orchard and wild hosts) within each geographic location separately, testing the effects of the different seasons, the two years of flight monitoring and the mean daily temperature. Trap captures were handled as longitudinal data and the respective analysis was chosen because the subjects (traps) create clusters with each observation (capture) corresponding to the same trap. The structure assumed for the clusters was auto-regressive of the order 1, with alpha = 0.532 and standard error = 0.068, regarding the model that included both locations, alpha = 0.324 with standard error = 0.074 for the model referring to the coastal area and last, alpha = 0.390 with standard error = 0.029 for the model that referring to the semi-highland area. The correlation between the within-subjects observations reduced exponentially as the distance between the two time instances increased. The selection of the respective correlation structure was based on the nature of the experiment itself, as the captures tend to be correlated more in adjacent instances, and on the Quasi Information Criteria (QIC). We used the log link function for the response variable “trap captures” (count data) and we modeled the variance using Poisson distribution. Parameter estimates were presented as Incidence Rate Ratios (IRR) with 95% confidence intervals (CI), which is the ratio of the number of trap captures in a group of interest to the number of trap captures of the group used as reference. IRRs greater than 1 indicate more trap captures for the group of interest while IRRs lower than 1 indicate more trap captures for the reference group. Estimated marginal means for the treatment and season factor were calculated with pairwise contrasts using Tukey adjustment for multiple comparisons.

Comparisons of cherry infestation rates were made by performing logistic regression using SPSS 26.0 (SPSS Inc., Chicago, IL, USA). The Generalized Linear Model with binomial distribution was built using infestation as the response variable, and the factors location, year of flight monitoring, treatment (conventional vs. unmanaged) and cultivar (early vs. late maturing cherry cultivar) as fixed effects. The model effects included all two-way interactions of the factors that significantly affected cherry infestation. Parameter estimates were presented as odds ratios (OR) with 95% confidence intervals (CI) using no infestation as the reference group.

Finally, possible effects of the above-mentioned factors on the number of adults emerging from the infested cherries were investigated by performing a Univariate Analysis of Variance. For the data modelling a Poisson Generalized Linear Model was used with a log link function. The model was built using the response variable of emerged adults and the fixed effects of location (Lehonia, Agia Fotini), year (2018, 2019), treatment (conventional and unmanaged cherry trees) and cultivar (early and late maturing cherry cultivar).

## 3. Results

### 3.1. Population Dynamics in the Two Geographic Locations

Seasonal patterns of adult *D. suzukii* captures in Lehonia and Agia Fotini are given in Figure 3 (see also Appendix A). More flies in total were captured at the semi-highland area Agia Fotini than at the coastal area Lehonia (IRR (95% CI) = 4.51 (2.49, 8.17), *p* < 0.001) (Table 3). Adult captures were higher during 2018 than 2019 and 2020 (IRR (95% CI) = 0.24 (0.18, 0.33), *p* < 0.001), adjusting for the geographic area (Figure 3). The mean daily temperature and the different seasons also significantly affected adult captures, in both areas and years of observation (Table 3).

Trap captures at the coastal area were recorded throughout the year with the exception of hot-dry summer months and exhibited two peaks: in spring and late-autumn/early-winter (Figure 3, Appendix A). During the first year of trap monitoring, captures in Lehonia were observed from the onset of the traps’ placement in March, continued throughout spring and reached the first peak early in summer (1 June 2018). Trap captures seized during summer 2018 and started building up once again during late-autumn reaching a second peak in late-October 2018 (Appendix A). Adult activity in Lehonia continued throughout autumn and winter 2018 until late spring of the following year, displaying the first peak of the new season on 4 May 2019 (Appendix A). As in the case of the previous year, trap captures ceased during the hot summer months of 2019 and resumed in mid-October 2019. Peaks were recorded during winter (on 7 December 2019 and on 11 January 2020).

Adversely, the insect’s population dynamics followed a different seasonal pattern at the semi-highland area (Figure 3, Appendix A). No trap captures were observed during most of the winter months and spring for both years. Population build-up in this area started early in June and peaked in autumn of both 2018 (Appendix A) and 2019 (Appendix A). Although seasonality of captures followed a similar pattern during both years, trapping levels in Agia Fotini were higher from mid-August until mid-November in 2018 than 2019 (Appendix A). The insect’s population trends in this area do not seem to follow a “double-peak pattern” as the one observed in Lehonia and in other areas with subtropical climate [33] or mild winters [26], but rather a single-peak period extending from summer to autumn. Very few or zero captures were recorded during winter and spring.

### 3.2. Population Dynamics in Different Types of Orchards at the Coastal Area Lehonia

Patterns of adult captures in unmanaged and conventional cherry orchards, as well as in an organic mixed farm and on wild hosts of Lehonia, during the two years of observation are given in Figure 4 (see also Appendix A). Adult captures were higher in unmanaged compared to conventionally managed cherry orchards (IRR (95% CI) = 0.39 (0.18, 0.87), *p* = 0.021), and the organic mixed farm (IRR (95% CI) = 0.07 (0.05, 0.11), *p* < 0.001), adjusting for year, season and temperature. No differences were observed in trap captures between unmanaged cherry orchards and wild hosts (Table 4). Additionally, season and mean daily temperature strongly affected adult captures, adjusting for the treatment and year, with lower captures in summer (IRR (95% CI) = 0.17 (0.12, 0.23), *p* < 0.001) than in spring, and with more captures in autumn (IRR (95% CI) = 3.17 (2.08, 4.84), *p* < 0.001) and winter (IRR (95% CI) = 1.97 (1.50, 2.57), *p* < 0.001) than in spring. No significant differences were observed in adult captures between the two years of observation (IRR (95% CI) = 0.99 (0.75, 1.33), *p* = 0.992) (Table 4, Figure 4).

Seasonal patterns of trap captures were different in the four types of orchards (Figure 5). Captures in unmanaged cherry orchards were higher during the cherry ripening period (March–April) than later in summer (Mean ratio _spring/summer_ = 5.93, *p* < 0.001). In conventionally treated cherry orchards capture rates were lower in spring than in summer (Mean ratio _spring/summer_ = 0.07, *p* = 0.009), and peaks were recorded in autumn, well after sweet-cherry harvest (Mean ratio _spring/autumn_ = 0.19, *p* < 0.001) (Figure 5). Adults were captured throughout autumn and winter until spring in the conventionally treated and unmanaged cherry orchard, despite the absence of fruit (Figure 5). Capture rates in the organic mixed orchard were constantly low during all seasons and peaked in autumn and winter (Mean ratio _spring/autumn_ = 0.09, *p* < 0.001, Mean ratio _spring/winter_ = 0.21, *p* < 0.001, Mean ratio _summer/autumn_ = 0.03, *p* < 0.001, Mean ratio _summer/winter_ = 0.08, *p* < 0.001). Traps placed on wild hosts captured lower number of adults than those placed in conventional sweet-cherry orchards during spring and summer (Mean ratio _cherries/wild hosts_ = 106.28, *p* < 0.001). Higher capture rates (though no significant) were recorded in wild hosts compared to conventional cherry orchards during winter (Mean ratio _conventional cherries/wild hosts_ = 0.70, *p* = 0.880). Adult captures in wild hosts were higher in winter compared to the rest of the year (Figure 5).

### 3.3. Population Dynamics in Different Types of Orchards at the Mainland Area Agia Fotini

Patterns of adult captures in unmanaged and conventional cherry orchards, and on wild hosts of Agia Fotini, during the two years of observation are given in Figure 6 (see also Appendix A). Adult captures were higher in unmanaged compared to conventionally managed cherry orchards (IRR (95% CI) = 0.62 (0.40, 0.97), *p* = 0.034) adjusting for year, season and temperature (Table 4). Adult captures were higher during spring 2018 to winter 2019 than during the same period of the following year (IRR (95% CI) = 0.17 (0.14, 0.20), *p* < 0.001) adjusting for treatment, season and temperature. Season strongly affected adult captures with more captures in summer (IRR (95% CI) = 60.70 (48.00, 76.70), *p* < 0.001), autumn (IRR (95% CI) = 115.00 (92.00, 143.00), *p* < 0.001) and winter (IRR (95% CI) = 18.50 (13.10, 26.10), *p* < 0.001) than in spring. Temperature fluctuation was also correlated with adult captures (IRR (95% CI) = 1.12 (1.09, 1.14), *p* < 0.001), (Table 4).

As in the case of the coastal area Lehonia, adult captures in Agia Fotini followed different seasonal patterns in sweet-cherry orchards than in wild hosts (Figure 7). Adult captures followed similar seasonal trends in conventionally-treated sweet-cherry orchards and abandoned/unmanaged cherry trees, with increased capture rates during summer, and even at the absence of cherries in autumn. Adult captures dropped dramatically during winter. Adult captures in wild hosts were lower than in sweet-cherry trees all through the fruiting season in summer (Mean ratio _unmanaged/wild hosts_ = 3.67, *p* < 0.001, Mean ratio _conventional/wild hosts_ = 1.66, *p* = 0.004) and significantly increased during winter (Mean ratio _conventional/wild hosts_ = 0.43, *p* = 0.012) (Figure 7).

### 3.4. Sweet-Cherry Infestation

The percentage of infested cherries sampled from early and late-ripening cherry cultivars of unmanaged, as well as of conventionally-treated cherry orchards in Lehonia and Agia Fotini during 2018 and 2019 is shown in Figure 8. Infestation rates were higher in Lehonia (38.1%) than in Agia Fotini (16.1%), adjusting for sampling year, cultivar and treatment (OR 95% CI) = 1.60 (1.07, 2.40), *p* = 0.023) (Table 5). Higher infestation rates were recorded in 2018 (36.4%) than in 2019 (20.9%), adjusting for location, cultivar and treatment (OR 95% CI) = 2.70 (1.80, 4.05), *p* < 0.001). Infestation rates in unmanaged cherry orchards were higher (46.875%) than in conventionally treated ones (13.92%), adjusting for location, cultivar and sampling year (OR 95%CI) = 6.08 (4.79, 7.73), *p* < 0.001). Moreover, fruit infestation rates were higher in late (33.22%) than early ripening cherry cultivars (22.09%), adjusting for sampling year, location and treatment (OR 95% CI) = 0.09 (0.06, 0.15), *p* < 0.001), (Table 5). The significant interaction between location and year shows that the reduction in sweet-cherry infestation levels during the second year of sampling was higher at the semi-highland area Agia Fotini. The interaction between location and cherry cultivar (*p* < 0.05) reveals that differences in infestation levels between early and late-ripening cherry cultivars were more pronounced in Agia Fotini than in Lehonia (Figure 8, Table 5).

The number of *D. suzukii* adults that emerged from infested cherries collected from early and late ripening cherry cultivars of Lehonia and Agia Fotini during 2018 and 2019 is given in Figure 9. More adults emerged from infested cherries sampled in 2018 (292 adults, 53.8%) than in 2019 (251 adults, 46.2%) (Table 5). Though statistically not significant, infested cherries from late ripening cultivars yielded more adults (299 adults, 55.1%) than those from early ripening ones (244 adults, 44.9%). It should be noted that the heavy infestation of late ripening cherry cultivars, of conventionally treated commercial orchards, in Agia Fotini during 2018 may account for the higher amount of adults emerging from the respective cherries (Figure 9). No other factors affected the number of *D. suzukii* adults emerging from infested cherries.

## 4. Discussion

The results of the current study demonstrate that *D. suzukii* is well adapted to the climatic conditions of both the coastal and the mainland (semi-highland) areas considered, confirming its wide thermal tolerance and its ability to survive and reproduce in numerous habitats [2,34]. Our survey also provides information regarding the susceptibility of early and late-ripening sweet-cherry cultivars.

Evidenced by the captures in traps, adult activity in the coastal area was prolonged from autumn until late spring of the following year and peaked twice (autumn and spring), during both years tested. The mean daily temperature of this area ranged from 10 °C to 20 °C during autumn, from 0 °C to 15 °C during winter and from 10 °C to 20 °C during spring. Moreover, during the hot summer months, the mean daily temperature was constantly above 25 °C, often exceeding 30 °C. On the other hand, the insect’s flight activity in the semi-highland area Agia Fotini was much shorter, lasted from summer until early winter of the following year, peaked in autumn and seized during the cold winter months. The mean daily temperature during autumn and spring in the respective area rarely exceeded the 10 °C–15 °C interval and often fell below 5 °C, while during winter it was never higher than 5 °C and often fell below 0 °C.

As shown previously, the optimum temperature for *D. suzukii* adult activity and immature development is 20 °C to 25 °C, with adult flies becoming inactive in temperatures below 8 °C–10 °C and over 28 °C–31 °C [35,36,37]. Furthermore, by developing a stage-specific population model for *D. suzukii*, Winman and co-authors [11] showed that environmental factors, with temperature being the most significant one, greatly affect the seasonal flight activity of adults. A temperature rise above 28 °C–30 °C, in the coastal area, probably leads to a sudden decline in adult flight activity, which becomes restored as soon as mean daily temperature drops below the 30 °C threshold [26,33,38]. Other factors that may account for the null adult trap captures during summer in the coastal area may be low survival in high temperatures [10], as well as migration to higher altitudes of the adjacent Pelion Mountain [25].

The delayed onset of adult flight in summer and the overall shorter period of adult activity in Agia Fotini are probably induced by the persisting low temperatures during winter and spring in the respective area. Likewise, in temperate areas such as Oregon and Wisconsin, trap captures ceased after periods of low or subzero temperatures [34]. Population trends in other areas with similar seasonal temperature ranges, such as Marion County in Oregon and Jan Joaquin County in California, exhibit either the single or the double peak pattern, respectively, as in the case of our data [26,34,39]. It seems that extreme high temperatures interrupt adult flight activity during summer (double peak pattern), while extreme cold winter temperatures delay population build-up during late spring and lead to a peak in summer–autumn (single peak pattern). The pause in flight activity during the cold winter months at the semi-highland and during the hot summer months at the coastal area do not preclude successful overwintering and flight re-activation when the climatic conditions are once again favorable [40].

Trap captures were significantly higher during the first year of flight monitoring compared to the second year, at both areas and especially at the semi-highland cherry growing area Agia Fotini. June of 2018 was characterized by heavy rainfalls and sporadic hail storms that destroyed the majority of late ripening sweet cherries of the semi-highland area. Sweet cherries in several parts of the area lost their commercial value, were not collected and remained on the cherry trees constituting an abundant oviposition substrate for *D. suzukii* and other Drosophilidae. Cherry harvest was performed at the coastal area in May–June 2018 as no damage was caused by rainfall. Interestingly, the insect’s flight activity persisted long after cherry harvest (late summer, autumn, winter) within cherry orchards in both areas. This probably indicates that factors such as the orchard’s microclimate, fallen or remaining cherry fruit and additional cherry trees secretions, proximity to alternate host and refuges provide a favorable environment for *D. suzukii* population survival and overwintering, as previously reported [4,26].

Higher adult captures were recorded during winter close to wild hosts (*Rubus* sp., *Pinus* sp.) of both areas, and within the organic mixed orchard of the coastal area, compared to the rest of the seasons. Cherry orchards of the semi-highland Agia Fotini consist the main cultivation of the area and often neighbor with forest or uncultivated plains with several available wild hosts of *D. suzukii*. Cherry orchards of the coastal Lehonia are located among various other fruit orchards, summer houses and farms (patchy landscape) with plenty of wild and ornamental vegetation intervening among them. Both landscapes facilitate the use of resources from crops, wild vegetation and ornamental plants in forests and field margins [41,42]. Wild hosts such as *Rubus* sp., *Solanum nigra*, and *Bryonia cretica*, that consist of representative species of the Mediterranean flora, as well as cultivated crops such as figs and pears, provide fruits in late summer and autumn, and may sustain *D. suzukii* populations during off-cherry season, facilitating its overwintering [4,39].

Late ripening cherry cultivars were infested at higher rates than early ripening ones in both areas tested and especially at the semi-highland Agia Fotini. Specifically, infestation rates of early ripening cherry cultivars of Agia Fotini had extremely low infestation rates even when cherries were collected from unmanaged/abandoned cherry trees. It appears that these cultivars are harvested before the insect’s population build-up and/or reproductive maturation takes place in summer. On the other hand, both early and late ripening cherry cultivars of the coastal area Lehonia are susceptible to infestation from *D. suzukii*. Unmanaged early and late ripening cherries in the coastal area were almost entirely (100%) or heavily (40% to 60%) infested, respectively. This drop in infestation levels could be attributed to the temperature rise and the patterns of adult population density in the area. Furthermore, the fact that mating frequencies, fertility and fecundity decrease in *D. suzukii* after exposure to high temperatures (>25 °C), may also account for the reduced infestation of late ripening cultivars in the coastal area [37]. Late-ripening cherry cultivars yielded more adults than early-ripening ones, indicating possible higher oviposition pressure that the respective cherry cultivars received due to increased population densities, as well as different susceptibility levels that could be related to skin firmness, Brix, color and other physicochemical characteristics [7,43].

In order to design a successful management plan for the control of *D. suzukii* in sweet cherry orchards, early detection is crucial, especially when the installed trap network includes alternative hosts, such as other fruit crops, wild and forest vegetation situated in proximity to the cherry orchard [42,44]. Flight monitoring using apple cider vinegar lure traps has been poorly correlated with the prediction of fruit infestation [45], and therefore it is not safe to solely rely on these data in order to design a pest management plan. However, information on the abundance and phenology of wild host species, coupled with the area’s climatic data, may assist the prediction of pest damage in specific production areas. Differences in susceptibility between different sweet cherry cultivars suggest, for example, that selection of early ripening cultivars when establishing a new orchard in a temperate area may chronologically avoid infestation by *D. suzukii*.

## 5. Conclusions

We conclude that *D. suzukii*, a pest of several high-value fruit, has adapted to both coastal and mainland areas of Greece where sweet cherries are cultivated and can induce great losses in sweet cherry production. Seasonal prevailing temperatures in each area and availability of wild hosts, adjacent to the cherry orchards, affect the insect’s population dynamics, by acting on adult flight activity and successful overwintering, respectively. Different population dynamics of this pest in each area may affect its potential to infest cherry cultivars with different ripening periods. Our data provide useful information for planning a successful management strategy against this pest in sweet cherry production areas of Greece and other Mediterranean countries.

## Figures and Tables

**Figure 1 insects-11-00621-f001:**
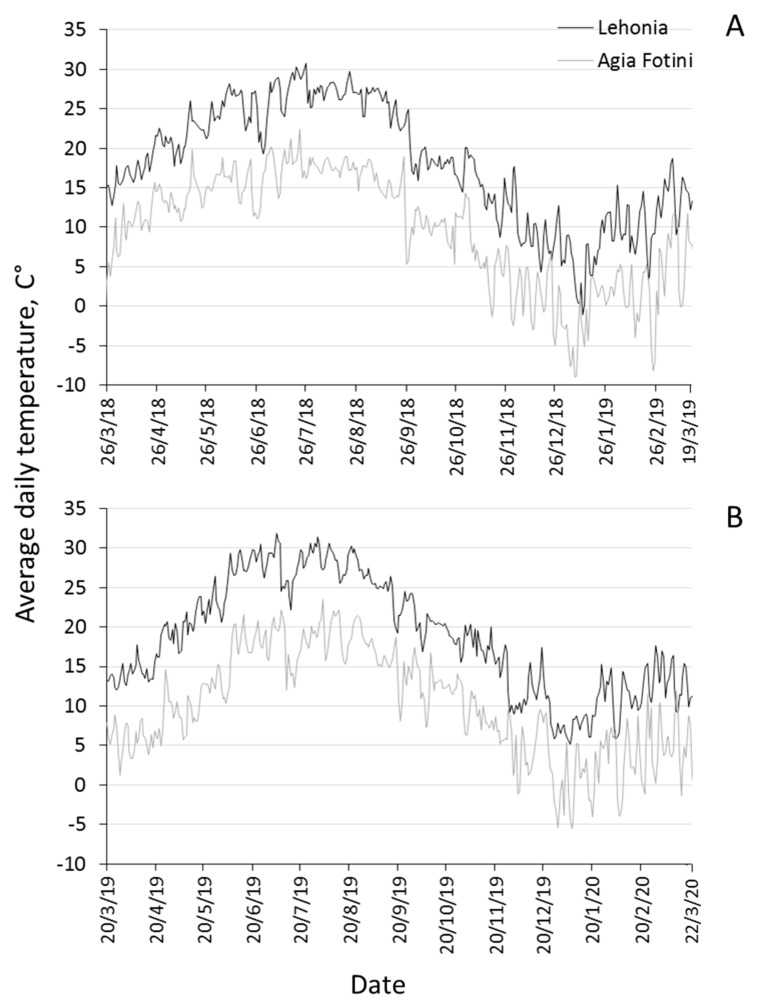
Seasonal patterns of daily temperature in Lehonia and Agia Fotini from 26 March 2018 to 19 March 2019 (**A**) and from 20 March 2019 to 22 March 2020 (**B**).

**Figure 2 insects-11-00621-f002:**
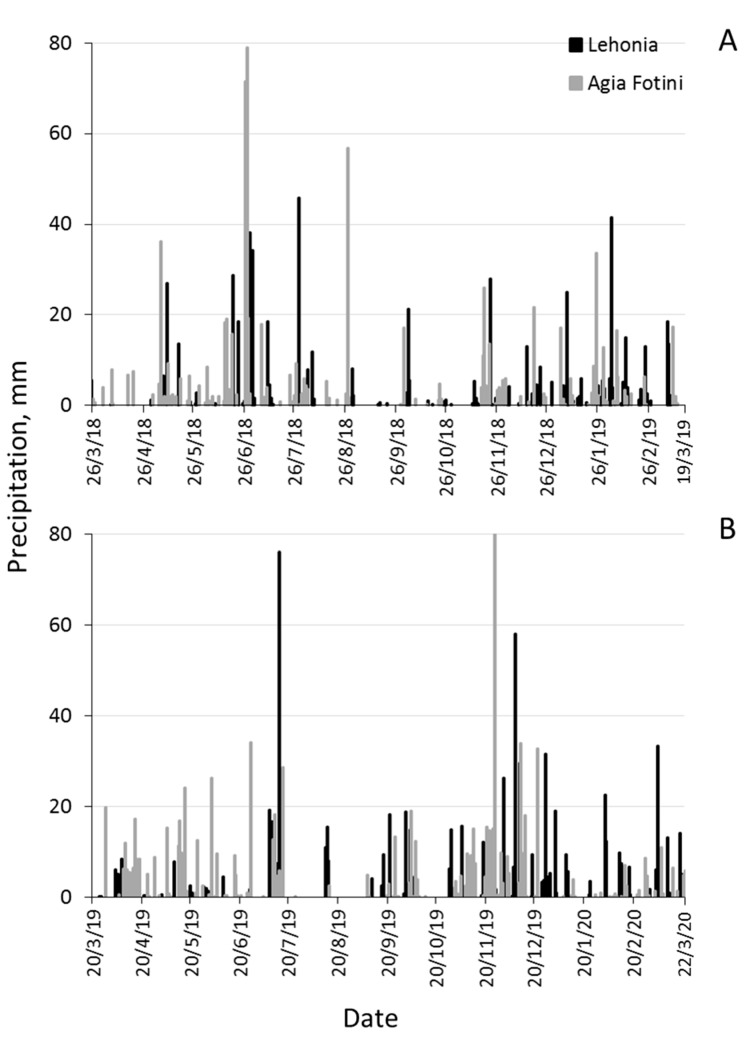
Seasonal patterns of daily precipitation levels in Lehonia and Agia Fotini from 26 March 2018 to 19 March 2019 (**A**) and from 20 March 2019 to 22 March 2020 (**B**).

**Figure 3 insects-11-00621-f003:**
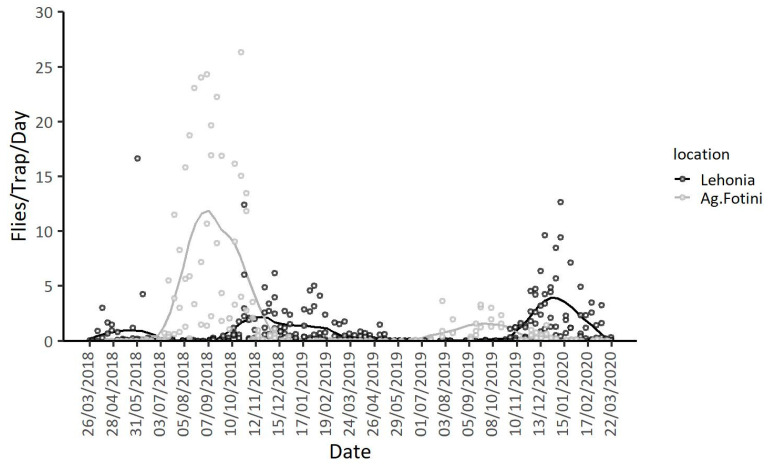
Adult captures per trap per day and LOESS curves (Locally Estimated Scatterplot Smoothing curves) for Lehonia and Agia Fotini from 26 March 2018 to 22 March 2020.

**Figure 4 insects-11-00621-f004:**
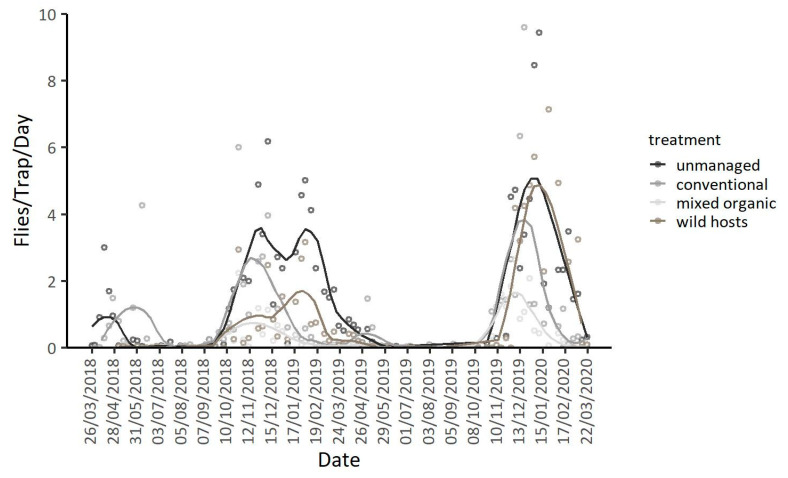
Adult captures per trap per day and LOESS curves (Locally Estimated Scatterplot Smoothing curves) for unmanaged and conventional cherry trees, an organic mixed orchard and wild vegetation in Lehonia from 26 March 2018 to 22 March 2020.

**Figure 5 insects-11-00621-f005:**
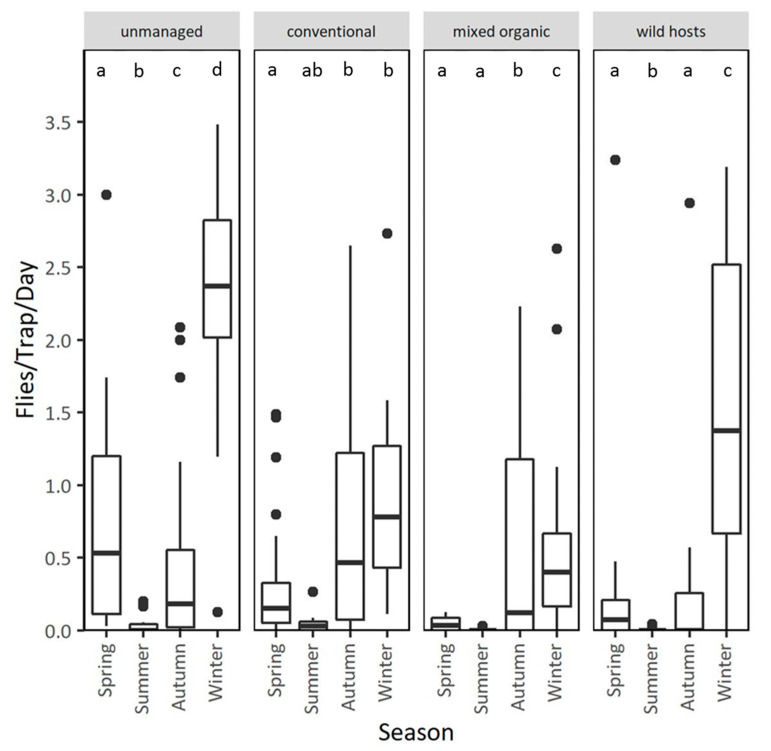
Adult captures in unmanaged and conventionally treated cherry trees, stone fruit trees of a mixed organic orchard and wild hosts–*Rubus* sp., of the coastal area Lehonia, grouped by season. Values within a chart followed by the same letter do not differ significantly (*p* > 0.05).

**Figure 6 insects-11-00621-f006:**
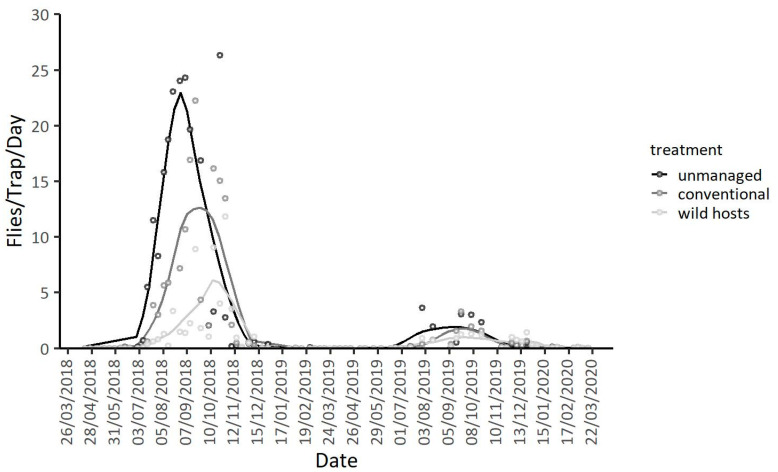
Adult captures per trap per day and smoothed LOESS curves for unmanaged, conventional cherry trees, and wild vegetation in Agia Fotini from 26 March 2018 to 22 March 2020.

**Figure 7 insects-11-00621-f007:**
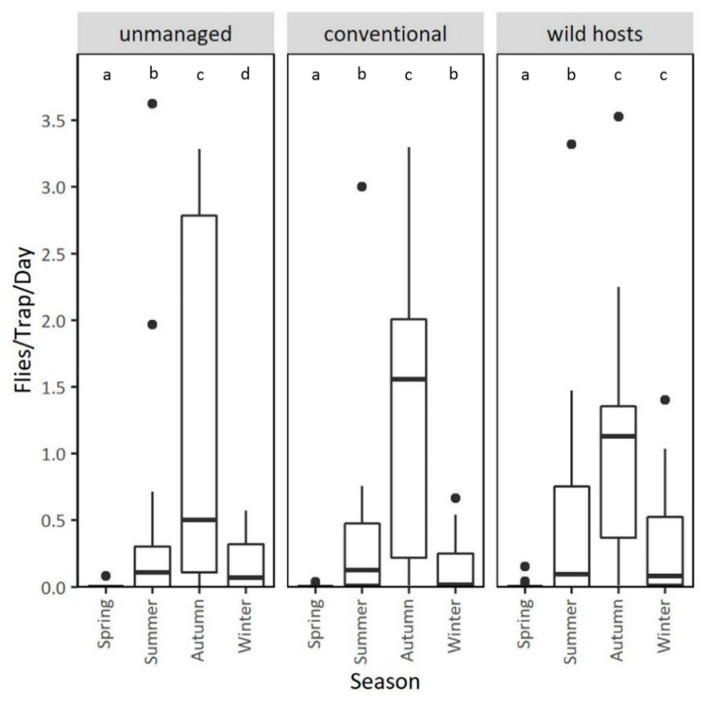
Adult captures in unmanaged and conventionally treated cherry trees, and wild hosts–*Rubus* sp., of the semi-highland area Agia Fotini, grouped by season. Values within a chart followed by the same letter do not differ significantly (*p* > 0.05).

**Figure 8 insects-11-00621-f008:**
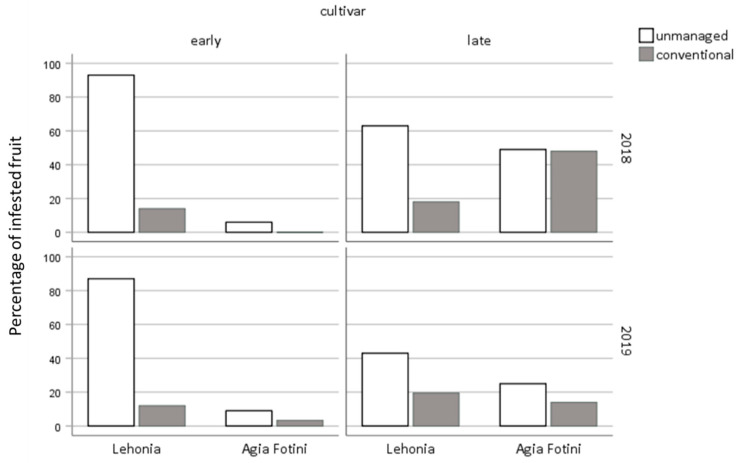
Sweet cherries infestation rates, sampled from early- and late-ripening cultivars of unmanaged and conventionally treated cherry orchards, located in Lehonia and Agia Fotini during 2018 and 2019.

**Figure 9 insects-11-00621-f009:**
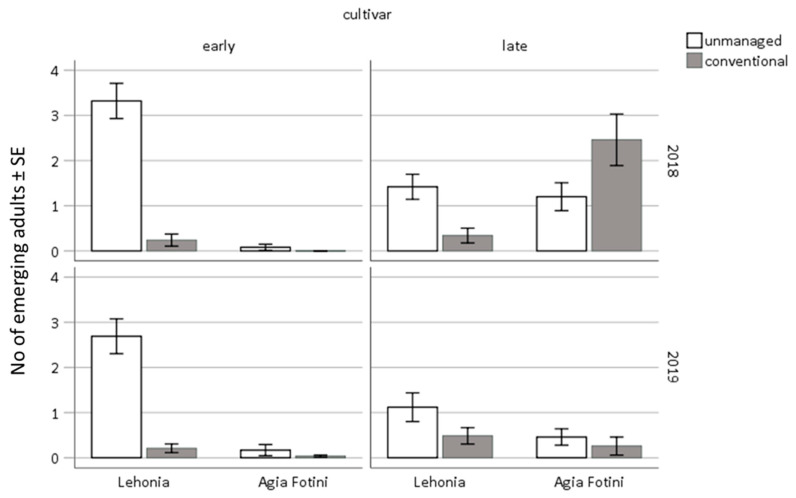
Adults emerging from field-infested sweet cherries, sampled from early- and late-ripening cultivars of unmanaged sweet-cherry trees and conventional commercial cherry orchards, located in Lehonia and Agia Fotini during 2018 and 2019.

**Table 1 insects-11-00621-t001:** Geographic locations, orchards, hosts and number of traps allocated for monitoring the flight activity of *Drosophila suzukii* adults in Pella and Magnisia Counties, during 2018 and 2019.

Location	Orchards	Lat. (North)	Long. (East)	Treatment	Number of Traps	Date of Trap Establishment
Lehonia ^1^	1	39°19′28.12″	23°2′57.53″	Unmanaged cherries	10	16 March 2018
				Wild hosts	2	26 April 2018
	2	39°19′27.99″	23°2′20.85″	Cherries	10	26 March 2018
				Wild hosts	2	26 April 2018
	3	39°19′37.93″	23°1′35.19″	Cherries	5	30 March 2018
	4	39°19′9.21″	23°1′34.53″	Stone fruit	2	26 April 2018
				Pome fruit	3	26 April 2018
				Wild host	1	26 April 2018
Agia Fotini ^2^	1	40°43′8.46″	22°0′20.63″	Conventional cherries	10	11 April 2018
				Unmanaged cherries	2	11 April 2018
				Wild hosts	2	11 April 2018
	2	40°42′0.10″	22°0′14.80″	Conventional cherries	10	11 April 2018
				Wild hosts	2	11 April 2018
	3	40°42′7.07″	22°0′25.64″	Conventional cherries	3	31 March 2019
				Wild hosts	2	31 March 2019
	4	40°41′30.56″	21°59′55.08″	Conventional cherries	3	31 March 2019
				Wild hosts	2	31 March 2019
	5	40°41′42.51″	22°0′1.00″	Conventional cherries	3	31 March 2019
				Wild hosts	2	31 March 2019
	6	40°42′55.97″	22°0′20.79″	Conventional cherries	3	31 March 2019
				Wild hosts	2	31 March 2019

^1^ Orchard No 1 was abandoned and unmanaged. Orchards No 2 and 3 were farmed conventionally and organic orchard No 4 included fruit trees and vegetables; ^2^ Cherry orchards were farmed conventionally. Insecticides used for the control of *D. suzukii* during the cherry fruiting seasons included neonicotinoids (i.e., acetamiprid) and sulfoxamines (i.e., sulfoxaflor). The maximum distance between orchards was 3.0–3.5 km. Unmanaged cherry trees and wild hosts were adjacent to the cherry blocks.

**Table 2 insects-11-00621-t002:** Sampling dates and amount of cherries collected from early and late maturing cherry cultivars of conventionally managed and abandoned/unmanaged trees in Lehonia and Agia Fotini during 2018 and 2019.

Location	Year	Ripening Period	Sweet-Cherry Cultivar	Treatment	Cherry Harvest/Sampling Dates	Number of Cherries
Lehonia	2018	Early	Sweet early	Conventional	5 May	100
			Early lory	Unmanaged	5 May	100
		Late	Precoce bernard	Conventional	14–20 May	100
			Precoce bernard	Unmanaged	14–20 May	100
	2019	Early	Sweet early	Conventional	11 May	200
			Early lory	Unmanaged	11 May	100
		Late	Precoce bernard	Conventional	1 June	200
			Precoce bernard	Unmanaged	28 May	100
Agia Fotini	2018	Early	Prime giant	Conventional	6 June	100
			Skeena	Unmanaged	24 June	100
		Late	Gesmestorfer	Conventional	8 July	100
			Gesmestorfer	Unmanaged	16 July	100
	2019	Early	Prime giant, Skeena	Conventional	16 & 26 June	300
			Prime giant	Unmanaged	16 June	100
		Late	Gesmestorfer	Conventional	18 July	100
			Gesmestorfer	Unmanaged	18 July	100

**Table 3 insects-11-00621-t003:** Variables of the generalized estimating equations GEEs with significant effects on the total trap captures of adult *Drosophila suzukii* in the two locations tested.

Factor	IRR (95% CI)	*p* Value
Intercept	0.64 (0.26, 1.53)	0.313
Location (ref: Lehonia)	3.28 (1.84, 5.85)	<0.001
Year (ref: first year)	0.24 (0.18, 0.33)	<0.001
Season (ref: spring)		
summer	5.65 (2.28, 14.00)	<0.001
autumn	9.43 (4.54, 19.60)	<0.001
winter	5.75 (2.77, 11.90)	<0.001
Mean daily temperature	1.06 (1.03, 1.09)	<0.001

**Table 4 insects-11-00621-t004:** Variables of the GEEs with significant effects on the trap captures of adult *Drosophila suzukii* in the coastal area Lehonia and in the semi-highland area Agia Fotini.

Factor	Lehonia		Agia Fotini	
	IRR (95% CI)	*p* Value	IRR (95% CI)	*p* Value
Intercept	25.30 (16.10, 39.90)	<0.001	0.17 (0.11, 0.28)	<0.001
Treatment (ref: unmanaged)				
conventional	0.39 (0.18, 0.87)	0.021	0.62 (0.40, 0.97)	0.034
wild hosts	0.44 (0.13, 1.52)	0.196	1.38 (0.63, 3.02)	0.427
mixed organic	0.07 (0.05, 0.11)	<0.001	-	-
Year (ref: first year)	0.99 (0.75, 1.33)	0.992	0.17 (0.14, 0.20)	<0.001
Season (ref: spring)				
summer	0.17 (0.12, 0.23)	<0.001	60.70 (48.00, 76.70)	<0.001
autumn	3.17 (2.08, 4.84)	<0.001	115.00 (92.00, 143.00)	<0.001
winter	1.97 (1.50, 2.57)	<0.001	18.50 (13.10, 26.10)	<0.001
Mean Temperature	0.91 (0.90, 0.93)	<0.001	1.12 (1.09, 1.14)	<0.001
Treatment * Season ^1^				
conventional * summer	82.00 (16.90, 398.00)	<0.001	0.73 (0.56, 0.95)	0.017
wild hosts * summer	0.69 (0.18, 2.67)	0.588	0.20 (0.10, 0.41)	<0.001
mixed organic * summer	2.25 (1.29, 3.91)	<0.001	-	-
conventional * autumn	1.68 (0.81, 3.48)	0.168	1.42 (1.06, 1.91)	0.020
wild hosts * autumn	0.40 (0.16, 0.99)	0.048	0.41 (0.17, 0.98)	0.045
mixed organic * autumn	3.65 (2.13, 6.25)	<0.001	-	-
conventional * winter	1.30 (0.62, 2.70)	0.490	1.69 (1.14, 2.52)	0.009
wild hosts * winter	1.66 (0.98, 2.82)	0.059	1.79 (1.10, 2.93)	0.019
mixed organic * winter	2.42 (1.64, 3.59)	<0.001	-	-

^1^ Asterisk (*) indicates interactions between different factors.

**Table 5 insects-11-00621-t005:** Variables of the GLMs with significant effects on the cherry infestation levels and the number of adult *Drosophila suzukii* emerging from infested cherries.

Dependent Variable	Factor	OR (95% CI)	*p* Value
Cherry infestation	Intercept	0.11 (0.08, 0.16)	<0.001
	Location (ref: Ag. Fotini)	1.60 (1.07, 2.40)	0.023
	Year (ref: second year)	2.70 (1.80, 4.05)	<0.001
	Treatment (ref: unmanaged)	6.08 (4.79, 7.73)	<0.001
	Cherry cultivar (ref: late)	0.09 (0.06, 0.15)	<0.001
	Location (Lehonia) * Year (first) ^1^	0.56 (0.34, 0.93)	0.024
	Location (Lehonia) * Cherry cultivar (early)	19.61 (11.26, 34.15)	<0.001
Dependent Variable	Factor	IRR (95% CI)	*p* value
Emerging adults	Intercept	2.53 (2.23, 2.87)	<0.001
	Location (ref: Ag. Fotini)	0.94 (0.83, 1.06)	0.324
	Year (ref: second year)	1.27 (1.15, 1.41)	<0.001
	Treatment (ref: unmanaged)	0.94 (0.84, 1.06)	0.309
	Cherry cultivar (ref: late)	1.19 (1.00, 1.25)	0.050

^1^ Asterisk (*) indicates interactions between different factors.

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
