# Peer review of "Population Dynamics of Drosophila suzukii in Coastal and Mainland Sweet Cherry Orchards of Greece"

_insects, 2020, doi:10.3390/insects11090621_

Round 1
Reviewer 1 Report
This is a well-written manuscript that has investigated the seasonality of SWD in two different climatic regions in Greece. The methods applied are appropriate and the fact that authors have collected data from two consecutive years make the results robust. I only had a few minor edits that were provided in the attached pdf file. I suggest this manuscript be accepted after those edits.

Author Response
Lines 11-26 “I do not see much difference between two abstracts. I believe the simple summary is not necessary”
Answer: The section “simple summary” is mandatory in Insects journal, and as clearly stated in the “Instructions for Authors” Section, “Submissions without a simple summary will be returned directly”. However, we agree with reviewer’s point that the two abstracts are similar and an effort has been made to simplify/ improve the “simple summary” and maintain it in the text.
Line 27: Matsumura (Diptera: Drosophilidae) has been added to the text.
Line 32: the word ‘adjacent’ is replaced by ‘neighboring’.
Line 51: the phrase ‘resulting in the maintenance of high population densities’ is now replaced by the phrase ‘which support high fly densities’.
Lines 99-100: the phrase ‘a great deal’ is now replaced by the word ‘tremendously’.
Line 121: the word ‘adjacent’ is replaced by ‘neighboring’
Figure 1: “Edit the last date on the x-axis to 19/3/19 instead of 19/3/20”
Answer: The date in x-axis is now corrected.
Figure 2: “Change to 19/3/19.”
Answer: The date in x-axis is now corrected.
Figure 4: “Use different colors for the points and lines so they are easier to read”.
Answer: Lines and points in Figures 3, 4 and 6 have been improved to increase readability.
Reviewer 2 Report
The subject of the manuscript is relevant, but you should make the corrections suggested below
|
Simple Summary Line 11-14 |
Drosophila suzukii or spotted-wing drosophila is a fruit fly of the same family as 11 the vinegar fly. Although the majority of Drosophila fruit flies are not able to infest intact, healthy 12 fruit, D. suzukii is a pest of several high-value soft and thin skin fruits such as strawberries and sweet cherries. |
Please delete this sentence is not important |
|
Line 49 |
thin skin fruits |
Use thin epicarp |
|
Line 52-53 |
blackberries, blueberries, raspberries, strawberries and sweet cherries |
Add scientific name |
|
Line 62 |
|
Include this paragraph There are still environmentally adequate areas with potential for D. suzukii occurrence in Oceania and Africa, although there has been no record of species on these continentes
Reference dos Santos LA, Mendes MF, Krüger AP et al (2017) Global potential distribution of Drosophila suzukii (Diptera, Drosophilidae). Plos One 12: e0174318. https://doi.org/10.1371/journal.pone.0174318 |
|
Line 265-259 |
|
correct: figure 3 is superimposing on table 3 |
|
Table 2 |
|
Include de Cultivar names |
|
Methodology |
|
It would be important to make an analysis with several abiotic factors, and not only with the temperature |
|
Discussion |
|
I suggest to consult the article below and to use mainly in the discussion https://link.springer.com/article/10.1007/s13744-019-00686-5 |
|
Conclusion |
|
Please be more specific |
Author Response
Lines 11-14: “Please delete the first two sentences of the ‘simple summary’ they are not important”
Answer: We greatly appreciate this suggestion by the Reviewer. However, because the simple summary is addressed to a lay audience that may have no background knowledge on the subject studied, we prefer to keep these two introductory sentences.
Line 49: “Use thin epicarp instead of thin skin fruits”
Answer: The sentence has been changed to “...several high-value soft fruit and fruit with thin epicarp” (line 48).
Lines 52-53: “blackberries, blueberries, raspberries, strawberries and sweet cherries - Add scientific name”
Answer: The scientific names of the above-mentioned fruit species are now added in the text (lines 48-50).
Line 62: “Include this paragraph: “There are still environmentally adequate areas with potential for suzukii occurrence in Oceania and Africa, although there has been no record of species on these continents REF: dos Santos LA, Mendes MF, Krüger AP et al (2017) Global potential distribution of Drosophila suzukii (Diptera, Drosophilidae). Plos One 12: e0174318.”
Answer: We thank the Reviewer for this suggestion. The respective sentence is now included in the text and reference is added (Lines 60-62). The numbering of references within the text and at the end of the document is also properly edited.
Figure 3: “correct figure 3. It is superimposing on table 3”
Answer: Position of Figure 3 is now corrected in the document (lines 268-269).
Table 2: “Include the cultivar names”
Answer: All the sweet-cherry cultivar names are now included in Table 2.
Methodology: “It would be important to make an analysis with several abiotic factors, and not only with the temperature”
Answer: We greatly appreciate the point raised by the Reviewer. Although temperature appears to mostly affect the phenology of D. suzukii adults, other abiotic factors, such as relative humidity, precipitation and extreme climatic conditions (heavy rain, hail), may also act on the flight activity. Investigating the effects of other abiotic factors on adult population dynamics and even overwintering is an excellent idea for future studies.
Discussion: “I suggest to consult the article below and to use mainly in the discussion https://link.springer.com/article/10.1007/s13744-019-00686-5”
Answer: We thank the Reviewer for this suggestion and we now include information from the suggested paper in the “Results” and “Discussion” Section (lines 264 and 394). The numbering of references within the text and at the end of the document is also properly edited.
Conclusion: “Please be more specific”
Answer: An effort has been made to improve the “Conclusion” Section of our draft and make clearer to the reader (lines 458-463)